# Deep Signature: Characterization of Large-Scale Molecular Dynamics

**Tiexin Qin[1], Mengxu Zhu[1], Chunyang Li[2], Terry Lyons[3], Hong Yan[1], Haoliang Li[1,*]**
City University of Hong Kong[1] & Chengdu Institute of Biological Products co. Ltd[2]
& University of Oxford[3]
{tiexinqin,mengxuzhu}2-c@my.cityu.edu.hk,lichunyang@sinopharm.com
tlyons@maths.ox.ac.uk, {ityan,haoliang.li}@cityu.edu.hk

## Abstract

Understanding protein dynamics are essential for deciphering protein functional mechanisms and developing molecular therapies. However, the complex high-dimensional dynamics and interatomic interactions of biological processes pose significant challenge for existing computational techniques. In this paper, we approach this problem for the first time by introducing Deep Signature, a novel computationally tractable framework that characterizes complex dynamics and interatomic interactions based on their evolving trajectories. Specifically, our approach incorporates soft spectral clustering that locally aggregates cooperative dynamics to reduce the size of the system, as well as signature transform that collects iterated integrals to provide a global characterization of the non-smooth interactive dynamics. Theoretical analysis demonstrates that Deep Signature exhibits several desirable properties, including invariance to translation, near invariance to rotation, equivariance to permutation of atomic coordinates, and invariance under time reparameterization. Furthermore, experimental results on three benchmarks of biological processes verify that our approach can achieve superior performance compared to baseline methods.

## 1 Introduction

Biological processes are fundamentally driven by the dynamical changes of macromolecules, particularly proteins and enzymes, within their respective functional conformation spaces. Typical examples of such processes include protein–ligand binding, molecule transport and enzymatic reactions, and modern computational biologists investigate their underlying functional mechanisms by molecular dynamics (MD) simulations (Dror et al., 2012; Lewandowski et al., 2015). Built upon density functional theory (Car & Parrinello, 1985), MD has demonstrated remarkable capability in providing accurate atomic trajectories in three-dimensional (3D) conformational space and consist agreement with experimental observations (Frenkel & Smit, 2023).

The computational analysis of MD data has been a subject of extensive research for decades, with the goal of characterizing systems from trajectory information. However, due to the main challenge posed by intricate interatomic interactions over large-scale systems across inconsistent timescales, many existing works resort to oversimplified setups that incorporate biophysical priors to analyze certain aspects of dynamics such as protein fluctuations, relaxation time, stability, and state transitions (Law et al., 2017; Qiu et al., 2023). More recently, empowered by the parallel processing ability of GPUs, machine learning especially deep learning sheds new light on this field as it can discretize macromolecules as particles distributed in a 3D voxel grid and automatically learn their relations in a data-driven fashion (Li et al., 2020a; Rogers et al., 2023). Parallel to this, surface modeling-based approaches have emerged, firstly utilizing mathematical models to restore protein surfaces and then applying deep learning to analyze the chemical and geometrical features of surface regions around binding sites (Gainza et al., 2020b; Zhu et al., 2021). Despite the great potential of these approaches in automatic drug discovery, their computational complexity would increase linearly with the number of time stamps when processing MD data, struggling with application to

---

*Corresponding author.

long-time simulations. Besides, these methods commonly build upon coarse grained dynamics for accelerating computation. Nevertheless, selecting an optimal coarse graining mapping strategy that effectively simplifies the representation of the system while preserving essential features remains an open research problem (Jin et al., 2022; Majewski et al., 2023).

Another limitation of current MD analysis methods is the deficient utilization of structural bioinformatics for the largely increased difficulty in handling high-order interatomic interactions during dynamic processes. However, such structural bioinformatics, manifested in various covalent and non-covalent bonds, plays a pivotal role in molecular design for its capability of propagating local perturbations to facilitate conformational dynamics and alter biological function (Tsai et al., 2009; Otten et al., 2018). An illustrative example would be dihydrofolate reductase, which has been widely studied as important antitumor and antibacterial targets for treating tuberculosis and malaria. There exist four common mutations that confer drug resistance to antibiotics, proceeding in a stepwise fashion. Among them, the P21L mutation acts in a dynamical loop region associated with long range structural vibrations of the protein backbone, rather than directly on the active sites as other mutations (Toprak et al., 2012). Therefore, ignoring such interatomic interactive dynamics facilitated by molecular structure for a critical protein and counting the effects of active sites solely can result in biased assessments of designed drugs. Nonetheless, since the integration of structural bioinformatics into MD analysis would further introduce at least quadratic complexity with system size, existing works have not yet investigated this crucial aspect, highlighting a critical gap in our ability to comprehensively analyze and predict molecular behavior in drug design and resistance studies.

To this end, we aim to develop a computationally efficient framework that incorporates the structural bioinformatics with coarse graining mapping for automatically analyzing protein trajectory dynamics. In particular, we first introduce a graph clustering module that learns to extract coarse grained dynamics by approximating soft spectral clustering. With the clustering assignment function implemented by a graph neural network and parameters learned automatically, we circumvent the need for manual selection of coarse graining mapping. Subsequently, we introduce a path signature transform module served as a feature extractor to characterize the interatomic interactive dynamics after coarse graining. Path signature is a mathematically principled concept that utilizes iterated integrals to describe geometric rough paths in a compact yet rich manner (Lyons, 2014), thus suitable for our tasks where molecular trajectories are highly sampled and non-smooth. After attaching with a task-specific differentiable classifier or regressor, we devise an end-to-end framework, named Deep Signature, for efficiently characterizing the complex protein dynamics. Notably, due to the existence of considerable random fluctuation in simulated trajectories, ideal features ought to maintain symmetry respecting certain geometric transformations. We provide theoretical analysis that our extracted features exhibit invariance to translation, near invariance to rotation, equivariance to permutation of atomic coordinates, and invariance under time reparametrization of paths. Finally, we target our task on predicting functional properties of proteins from MD data, a fundamental task for developing novel drug therapies. We consider three benchmarks including gene regulatory dynamics, epidermal growth factor receptor (EGFR) mutation dynamics, and G protein-coupled receptors (GPCR) dynamics for performance evaluation. The contributions of our paper are as follows:

- We develop Deep signature, the first computationally efficient framework that characterizes the complex interatomic interactive dynamics of large-scale molecules.
- We theoretically demonstrate that our approach preserves symmetry under several geometrical transforms of atomic coordinates in 3D conformational space. Additionally, our method remains invariant under time reparameterization.
- We provide empirical results to show that our Deep Signature model achieves superior performance compared to other baseline methods on gene regulatory dynamics, EGFR mutation dynamics and GPCR dynamics.

## 2 RELATED WORKS

**Molecular Representation.** Encoding essential structural characteristics and biochemical properties into molecular representations is a long-standing research field in molecular biology, with wide applications in various drug discovery processes including virtual screening, similarity-based compound searches, target molecule ranking, drug toxicity prediction, and *etc*. (Li et al., 2024). One of the most widely used categories of molecular representation is two-dimensional fingerprints that extract the substructure, topological routes and circles solely from molecular connection tables. Owing to their ease of generation and usage, these 2D fingerprints are still extensively utilized as input

for machine learning algorithms in modern drug discovery applications (Gao et al., 2020). In recent years, there has been a surge in the development of 3D structure-based molecular representations for their much more fine-grained characterization of interatomic interplay in 3D conformational space. Surface modeling-based approaches leverage mathematical models to restore protein surfaces and encode geometrical and chemical features of surface regions around binding sites into representations, exhibiting great potential in automatic drug discovery (Gainza et al., 2020a; Zhu et al., 2021). In addition to surface-based representations, significant efforts have been devoted to learning accurate deep learning force fields for accelerating MD simulation (Schütt et al., 2017a; Batatia et al., 2022; Batzner et al., 2022). While these methods can be adapted for molecular property prediction, they are limited to generating representation for a static frame, thus not suitable for our tasks where interframe interactions along timescales are crucial for understanding protein function.

The current investigation into characterizing molecular dynamics, especially interatomic interaction dynamics, remains limited, with only a few studies close to ours. Among them, Endo et al. (2019), Yasuda et al. (2022) and Mustali et al. (2023) are unsupervised methods that build local dynamics ensembles for pre-specified atoms and inspect each atom's contribution independently. Li et al. (2022) converts MD conformations into images and applies convolutional neural networks to identify diverse active states. Sun et al. (2023) models protein dynamics by representing protein surfaces using implicit neural networks without requiring explicit surface representations. Nevertheless, these approaches cannot capture subtle interatomic interactions along atomic pathways for dissecting protein function, nor is the complexity of their representations independent of time stamps.

**Coarse Graining (CG).** CG is a widely adopted technique with the objective of preserving the crucial characteristics and dynamics inherent to a molecular system. This is achieved by grouping sets of atoms into CG beads, thereby enabling high-throughput MD simulations over larger time and length scales. Existing CG methods can be broadly categorized into two types: chemical and physical intuition-based approaches and machine learning-based approaches. The methods of first type construct the CG mapping by incorporating various biochemical properties derived from expert knowledge, for example, mapping each elaborately constructed cluster of four heavy (nonhydrogen) atoms into a single CG bead (Marrink & Tieleman, 2013) or simply assigning one CG bead centered at the $\alpha$-carbon for each amino acid (Ingólfsson et al., 2014). Machine learning-based methods can rapidly learn accurate potential energy functions for reduced structures of MD by training on large databases. Recent advancements in this area include multiscale coarse graining that optimizes to maximize a variational force matching score (Wang & Gómez-Bombarelli, 2019), relative entropy minimization (Thaler et al., 2022), and spectral graph approaches that account for structural typologies of proteins (Webb et al., 2018; Li et al., 2020b). However, their transferability to unseen molecules remains suspicious, and the representation capability for complex macromolecular systems without increasing dimension and complexity is still underexplored (Khot et al., 2019).

## 3 METHODOLOGY

### 3.1 PROBLEM FORMULATION

Consider various molecular systems $S^{(k)}$ that are distinct in their molecular behaviors. An MD simulation trajectory on $S^{(k)}$ provides the trajectories for all $N_k$ atoms constituting the molecules, and can be represented as a sequence of snapshots $\mathbf{X}_{1:T_k}^{(k)} = \{\mathbf{X}_1^{(k)}, \mathbf{X}_2^{(k)}, \ldots, \mathbf{X}_{T_k}^{(k)}\}$. Here, $\mathbf{X}_t^{(k)} \in \mathbb{R}^{N_k \times 3}$ indicates the atomic positions in 3D conformational space at time step $t$ for $t \in \{1, \ldots, T_k\}$ and $T_k$ is the total number of frames. To describe the structure of molecules within $S^{(k)}$, we define a molecular graph $\mathcal{G}^{(k)} = \{\mathcal{E}^{(k)}, \mathcal{V}^{(k)}\}$, $|\mathcal{V}^{(k)}| = N_k$, where $\mathcal{V}^{(k)}$ is the node set corresponding to the atoms and $\mathcal{E}^{(k)}$ is the edge set corresponding to chemical bonds. The adjacency matrix of $\mathcal{G}^{(k)}$ is represented by $\mathbf{A}^{(k)} \in \mathbb{R}^{N_k \times N_k}$, with $\mathbf{A}_{i,j}^{(k)} = 1$ if $v_i, v_j \in \mathcal{V}^{(k)}$ and $(v_i, v_j) \in \mathcal{E}^{(k)}$. For the molecular property prediction task, we have access to MD trajectories from $K$ molecular systems, each endowed with property labels denoted as $\{(\mathbf{X}_{1:T_k}^{(k)}, y^{(k)})\}_{k=1}^K$. The objective is to train algorithms to accurately predict the property label when provided with an previously unseen MD trajectory $\mathbf{X}_{1:T_k}$.

### 3.2 DEEP SIGNATURE

Our proposed method, referred to as Deep Signature, consists of a deep spectral clustering module that uses GNNs to extract coarse grained dynamics from raw MD trajectories, a path signature trans-

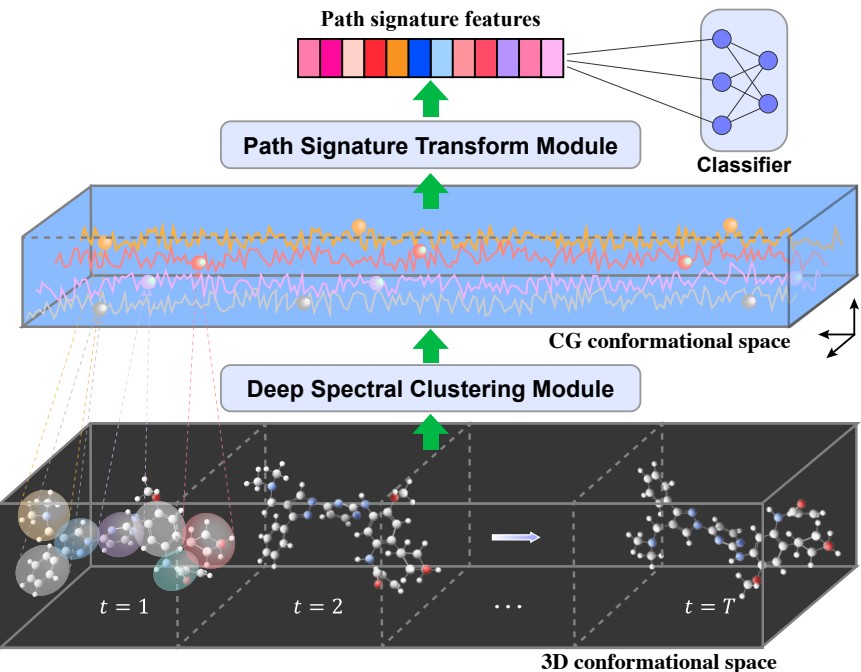

Figure 1: An overview of our proposed Deep Signature method.

form module that collects iterated integrals to characterize interatomic interactions along pathways, and a classifier to enable property prediction. The overall architecture is illustrated in Fig. 1.

**Deep spectral clustering with GNNs.** Given the MD trajectory $\mathbf{X}_{1:T}$ and molecular graph $\mathcal{G}$ for a molecular system $S$, we start with extracting the reduced trajectory $\tilde{\mathbf{X}}_{1:T}$ by coarse gaining $\mathbf{X}_{1:T}$ using deep spectral clustering. Specifically, we first obtain node representations via GNN layers as

$$\mathbf{H}^l = \sigma(\tilde{\mathbf{D}}^{-1/2}\tilde{\mathbf{A}}\tilde{\mathbf{D}}^{-1/2}\mathbf{H}^{l-1}\mathbf{W}_{\text{GNN}}^{l-1}), \tag{1}$$

where $\mathbf{H}^l$ denotes the node feature matrix at the $l$-th layer, $\mathbf{H}^0 = \mathbf{X}_{1:T}$, $\tilde{\mathbf{A}} = \mathbf{A} + \mathbf{I}$ is the adjacency matrix $\mathbf{A}$ plus the identity matrix $\mathbf{I}$, $\tilde{\mathbf{D}}$ is the degree matrix of $\tilde{\mathbf{A}}$, $\mathbf{W}_{\text{GNN}}^{l-1}$ are the learnable parameters of GNNs, and $\sigma$ is a nonlinear activation function. After then, we compute the cluster assignment matrix $\mathbf{Q}$ for the nodes using a multi-layer perceptron (MLP) with softmax on the output layer

$$\mathbf{Q} = Softmax(\mathbf{W}_{\text{MLP}}\mathbf{H}^l + \mathbf{b}), \tag{2}$$

where $\mathbf{W}_{\text{MLP}}$ and $\mathbf{b}$ are trainable parameters of the MLP. For the assignment matrix $\mathbf{Q} \in \mathbb{R}^{N \times M}$, where $M \ll N$ specifies the number of clusters, each row of it represents the node's probability of belonging to a particular cluster. We employ the normalized-cut relaxation (Bianchi et al., 2020) as our clustering objective for minimization

$$\mathcal{L}_u = -\frac{Tr(\mathbf{Q}^T\tilde{\mathbf{A}}\mathbf{Q})}{Tr(\mathbf{Q}^T\tilde{\mathbf{D}}\mathbf{Q})} + \left\|\frac{\mathbf{Q}^T\mathbf{Q}}{||\mathbf{Q}^T\mathbf{Q}||_F} - \frac{\mathbf{I}_M}{\sqrt{M}}\right\|_F, \tag{3}$$

where the first term promotes strongly connected components to be clustered together, while the second term encourages the cluster assignments to be orthogonal and have similar sizes.

Upon leveraging $\mathbf{Q}$ from Eq. (2) for clustering, the corresponding reduced feature embedding matrix $\mathbf{H}'$ and adjacency matrix $\mathbf{A}'$ can be derived as follows

$$\mathbf{H}' = \mathbf{Q}^T\mathbf{H}; \quad \mathbf{A}' = \mathbf{Q}^T\tilde{\mathbf{A}}\mathbf{Q}. \tag{4}$$

Since our model takes the sequence $\mathbf{X}_{1:T}$ as input, $\mathbf{H}'$ inherently maintains the temporal order in the form of $\mathbf{H}'_{1:T}$. We utilize another MLP with the parameters $\mathbf{W}'_{\text{MLP}}$ and $\mathbf{b}$ to map $\mathbf{H}'_{1:T}$ back into a reduced conformational space with the resulting dynamics and ground truth dynamics expressed as

$$\tilde{\mathbf{X}}_{1:T} = \mathbf{W}'_{\text{MLP}}\mathbf{H}'_{1:T} + \mathbf{b}'; \quad \tilde{\mathbf{X}}_{1:T}^{\text{pool}} = \mathbf{Q}^T\mathbf{X}_{1:T}. \tag{5}$$

To ensure the fidelity of the coarse grained dynamics towards the original high-dimensional system, we further introduce a temporal consistency constraint, defined through a mean absolute error loss function with the form

$$\mathcal{L}_t = \frac{1}{T}\sum_{i=1}^{T}\left|\tilde{\mathbf{X}}_i - \tilde{\mathbf{X}}_i^{\text{pool}}\right|. \tag{6}$$

**Path signature transform.** We now adopt the path signature method to characterize the interatomic temporal interactions among the coarse grained dynamics $\tilde{\mathbf{X}}_{1:T} \in \mathbb{R}^{T \times 3M}$. The basic idea of path signature is that, for a multidimensional continuous path, we can construct an ordered set consisting of all possible path integrals and combinations involving the path integrals among various individual dimensions as a comprehensive representation for this path (Lyons, 2014). Striving for a more precise definition, consider our coarse grained trajectory $\tilde{\mathbf{X}}_{1:T}$ with $(\tilde{\mathbf{X}}_t^1, \tilde{\mathbf{X}}_t^2, \ldots, \tilde{\mathbf{X}}_t^{3M})$ for $t \in \{1, \ldots, T\}$, let us define $\widehat{\mathbf{X}} : [1, T] \to \mathbb{R}^{3M}$ as a piecewise linear interpolation of $\tilde{\mathbf{X}}_{1:T}$ such that $\widehat{\mathbf{X}}_t = \tilde{\mathbf{X}}_t$ for any $t \in \{1, \ldots, T\}$, and a sub-time interval $[r_i, r_{i+1}]$ corresponding to a time partition of $[1, T]$ with $1 = r_1 < r_2 < \cdots < r_\tau = T$. The depth-$D$ signature transform of $\widehat{\mathbf{X}}$ over the interval $[r_i, r_{i+1}]$ is the vector defined as

$$\text{Sig}_{r_i, r_{i+1}}^D(\tilde{\mathbf{X}}) = \left(1, \left\{S(\tilde{\mathbf{X}})_{r_i, r_{i+1}}^j\right\}_{j=1}^{3M}, \ldots, \left\{S(\tilde{\mathbf{X}})_{r_i, r_{i+1}}^{j_1, \ldots, j_d}\right\}_{j_1, \ldots, j_d=1}^{3M}\right), \tag{7}$$

where for any $(j_1, \ldots, j_d) \in \{1, \ldots, 3M\}^D$,

$$S(\tilde{\mathbf{X}})_{r_i, r_{i+1}}^{j_1, \ldots, j_d} = \int \cdots \int_{1 < t_1 < \ldots < t_d < T} d\widehat{\mathbf{X}}_{t_1}^{j_1} \ldots d\widehat{\mathbf{X}}_{t_d}^{j_d}. \tag{8}$$

After that, we take the logarithm of the signature transform features presented in Eq. (7) to eliminate redundant elements according to the shuffle product identity (Lyons et al., 2007) and acquire some minimal collection of the stream over a time interval. Specifically, given the vector space described by depth-$D$ signature in formal power series form as

$$\text{Sig}_{r_i, r_{i+1}}^D(\tilde{\mathbf{X}}) = \sum_{d=0}^{D} \sum_{j_1, \ldots, j_d \in \{1, \ldots, 3M\}} S(\tilde{\mathbf{X}})_{1,T}^{j_1, \ldots, j_d} e_{j_1} \ldots e_{j_d}, \tag{9}$$

The depth-$D$ log-signature transform corresponds to taking the formal logarithm of Eq. (9), which can be expressed as

$$\text{LogSig}_{r_i, r_{i+1}}^D(\tilde{\mathbf{X}}) = \sum_{n=1}^{3M} S(\tilde{\mathbf{X}})_{r_i, r_{i+1}}^n e_n + \sum_{1 \le n < m \le 3M} \frac{1}{2}(S(\tilde{\mathbf{X}})_{r_i, r_{i+1}}^{n,m} - S(\tilde{\mathbf{X}})_{r_i, r_{i+1}}^{m,n})[e_n, e_m] + \ldots$$
$$\tag{10}$$

where $[e_n, e_m]$ is the Lie bracket (Reizenstein, 2017) defined by $[e_n, e_m] = e_n \times e_m - e_m \times e_n$. Generally, log-signature possesses a lower dimension compared to the original signature feature while carrying exactly the same information. Through this transform, the interatomic dynamic correlations in 3D space are embedded as geometric areas into features. The resulting signature sequence, denoted as $(\text{LogSig}_{r_1, r_2}^D(\tilde{\mathbf{X}}), \text{LogSig}_{r_2, r_3}^D(\tilde{\mathbf{X}}), \ldots)$, can be regarded as a discretization of dynamic data. In this work, we further utilize LSTM to tackle the nonlinear interactions between them.

**Classifier.** Finally, we introduce a classifier $f$ implemented as a two-layer MLP which outputs the predicted label $\hat{y} = f(\text{LogSig}_{1,T}^d(\tilde{\mathbf{X}}))$ for molecular property prediction. To establish the classification objective function, we leverage cross-entropy loss defined as follows

$$\mathcal{L}_c = -y \log(\hat{y}) - (1 - y) \log(1 - \hat{y}), \tag{11}$$

where $y$ is the ground truth label for the system and $\hat{y}$ is the predicted label generated by our model.

By combining the loss terms from Eqs. (3), (6) and (11), we arrive at the overall loss function for training our Deep Signature model as

$$\mathcal{L} = \lambda_1 \mathcal{L}_u + \lambda_2 \mathcal{L}_t + \lambda_3 \mathcal{L}_c. \tag{12}$$

Here, $\lambda_1$, $\lambda_2$ and $\lambda_3$ are scaling parameters to balance the contributions of these three loss terms.

It is worth noting that, recent research has delved into the integration of path signatures with GNNs for spatial-temporal modeling, and marked significant strides in traffic forecasting scenarios (Choi & Park, 2023; Wang & Zang, 2024). These approaches utilize path signatures to process temporal information for each node individually at a low level, while employing GNNs empowered neural differential equation at a higher level to ensure the preservation of spatial topological consistency during dynamic forecasting. Nonetheless, they fall short in characterizing the temporal interactions among different nodes, making them unsuitable for molecular property prediction tasks where the interactions among dominant kinetic pathways play an essential role in protein function (Araki et al., 2021; Mustali et al., 2023).

### 3.3 EQUIVARIANCE & INVARIANCE OF LOG-SIGNATURE

Physical properties often exhibit well-defined symmetry characteristics, and integrating such equivariance into our learned feature space can enhance interpretability and mitigate learning difficulty. In the subsequent analysis, we rigorously examine the equivariance of our deep signature in relation to various geometric transformations including translation, rotation, and permutation on atomic coordinates, as well as reparameterization over time. The complete proofs are provided in Appendix A.

**Translation invariance.** The coarse-grained dynamics acquired by a linear mapping in Eq. (5) demonstrate that $\tilde{\mathbf{X}}_{1:T}$ maintain equivariance with respect to translations on input trajectory $\mathbf{X}_{1:T}$. Besides, since the path signature is composed of iterated integrals, it inherits the property of translation invariance. As a result, log-signature features are translation-invariant for the input trajectory.

**Rotation invariance.** When a rotation is applied to the 3D conformational space, an equivalent rotation exists in the coarse grained conformational space, indicating the equivariance of coarse grained dynamics $\tilde{\mathbf{X}}_{1:T}$ to rotations of the input trajectory $\mathbf{X}_{1:T}$. Furthermore, while research on rotation invariance of path signatures for higher depth and multi-dimensional paths is still limited (Diehl, 2013), we demonstrate that the majority of elements constituting our Deep Signature features are rotation-invariant, particularly in large-scale systems like the molecular systems studied in this work.

**Permutation equivariance.** GCNs that aggregate contributions from neighboring atoms are invariant to permutations of those atoms, which implies that our coarse-grained dynamics are equivariant to permutations of atom indices. Additionally, the multi-indices used to index the iterated integrals in a signature are sorted in ascending order during implementation, ensuring that our signature features remain equivariant to permutations. This property is essential for facilitating the interpretability of our method, as it allows us to trace the dominant kinetic pathways that contribute to protein function.

**Time-reparametrization invariance.** Path signature possesses a powerful property that it remains invariant under time-reparameterization of the underlying stream (Lyons, 2014). This property substantially decreases the complexity of certain challenges by eliminating dependence on the sampling rate while preserving all other relevant information within the stream. Furthermore, it enhances the robustness of our method against deviations in molecular dynamics occurring over time scales.

### 3.4 IMPLEMENTATION

**Architecture.** Our introduced framework consists of three fundamental modules: a deep spectral clustering module, a path signature transform module and a classifier. We adopt a hierarchical pooling architecture to implement the deep spectral clustering module, which is schematically depicted in Fig. 2. As shown, it contains a stack of $L$ graph pooling layers with each layer consisting of a GCN layer for obtaining node embeddings and an MLP layer for cluster assignment, such that it would gradually coarsen the dynamics from the atomic level. The last GCN layer and MLP take the node feature matrix and adjacency matrix processed by the $L$-th graph pooling layer as input and output the final coarse grained dynamics $\tilde{\mathbf{X}}_{1:T}$.

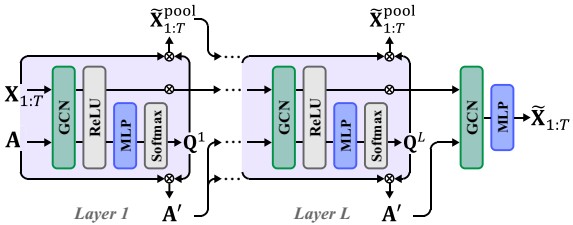

Figure 2: The architecture of deep spectral clustering module.

The implementation of the path signature transform module is illustrated in Fig. 3. It takes $\tilde{\mathbf{X}}_{1:T}$ as input, which is first partitioned into equally spaced intervals. The resulting segments are subsequently processed through the LogSig($\cdot$) function to extract log-signature features. This is implemented based on Signatory[1], a Python package that facilitates differentiable computations of the signature and log-signature transforms on both CPU and GPU. The resulting signature sequence is then fed into

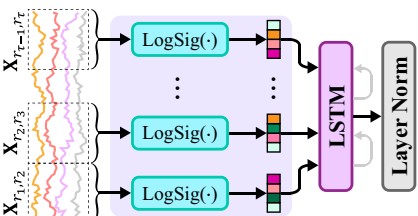

Figure 3: The architecture of path signature transform module.

---

[1] https://github.com/patrick-kidger/signatory

Table 1: Comparisons of classification performance on gene regulatory dynamics. Results are averaged over 5 runs.

| Method | Accuracy | Recall |
|--------|----------|--------|
| Head | $55.28_{\pm 10.56}$ | $30.65_{\pm 16.23}$ |
| Tail | $55.36_{\pm 10.72}$ | $30.71_{\pm 16.36}$ |
| Head & Tail | $67.92_{\pm 15.17}$ | $35.80_{\pm 9.27}$ |
| GraphLSTM | $96.40_{\pm 0.89}$ | $93.14_{\pm 1.20}$ |
| GraphTrans | $86.00_{\pm 6.67}$ | $79.36_{\pm 16.37}$ |
| Deep Signature | $99.12_{\pm 0.82}$ | $98.65_{\pm 0.23}$ |

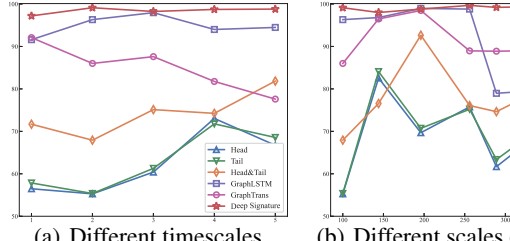

(a) Different timescales     (b) Different scales of system

Figure 4: The validation of scalability of our approach towards systems with various timescales (in a) and different number of atoms (in b).

an LSTM layer to capture their interactions and followed by layer normalization (Ba et al., 2016) to standardize the feature values. For the classifier, we utilize a two-layer MLP together with a sigmoid activation function to enable property prediction. It is noteworthy that the overall architecture is fully differentiable, thus we can leverage off-the-shelf optimization techniques to train our model in an end-to-end fashion.

**Cross validation and independent test.** We employ a five-fold cross-validation strategy for training our model. In contrast to conventional random division, we take the temporal nature of the trajectory data into consideration for data partition. Specifically, each trajectory is divided into 5 groups with the same time interval according to its temporal order. Subsequently, we further partition the data within each group into five folds. The validation set is constructed by selecting one fold from each group and is employed for model selection, while the remaining four folds within each group are gathered to form the training set. This process is repeated five times sequentially, resulting in the creation of the five-fold cross-validation dataset. Moreover, for each running, we evaluate the prediction accuracy of our method on an independent unseen test set and report the averaged results.

## 4 EXPERIMENTS

In this section, we conduct experiments on representative molecular dynamic systems to evaluate the effectiveness of our proposed method. We begin with a synthetic dataset that reports gene regulatory dynamics (Gao et al., 2016). We then assess the performance on two large-scale MD simulation datasets, including epidermal growth factor receptor (EGFR) mutant dynamics (Zhu et al., 2021) and G protein-coupled receptors (GPCR) dynamics (Rodríguez-Espigares et al., 2020). More details on dataset construction can be found in Appendix B.

### 4.1 EXPERIMENTS ON GENE REGULATORY DYNAMICS

For gene regulatory dynamics, we generate 100 trajectories that describe the interactive dynamics between genes and transcription factors. These dynamics are categorized into degradation type or dimerization type, with an equal number of trajectories for each type. The systems are simulated over a period of 2s and with a time interval of $0.004s$, resulting in 500 frames per trajectory. Besides, each system encompasses 100 nodes with randomly generated Power-law network structure to describe their relationships. We implement the deep spectral clustering module in our approach using two graph pooling layers that first coarsen the dynamics into 60 nodes and then into 30 nodes. The coefficients of loss terms are set as $\lambda_1 = 1$, $\lambda_2 = 0.01$, and $\lambda_3 = 10$. We optimize our model using Adam with an initial learning rate of 5e-4 and a weight decay of 1e-4. For comparison, we consider several baseline approaches which aggregate nodes' dynamics by averaging without taking into account their interactions. These methods incorporate a deep spectral clustering model with an MLP to process the first frame only (Head), the last frame only (Tail), the first and last frame (Head & Tail), and all frames with an additional LSTM layer (GraphLSTM) or Transformer (GraphTrans) to aggregate nodes' temporal dynamics. The overall comparison results with baseline methods are presented in Table 1.

From the table, we observe that our Deep Signature method achieves the best performance with a clear margin over baseline approaches. Besides, compared to GraphLSTM which aggregates atomic dynamics by averaging them directly, our Deep Signature exhibits a substantial performance boost

Table 2: Comparison of different methods for the classification results on the EGFR dynamics.

| Method | Dihedral angle | Cα-dihedral angle | Head | Tail | Head & Tail | GraphLSTM | GraphTrans | Deep Signature |
|---|---|---|---|---|---|---|---|---|
| **Accuracy** | $67.47_{\pm0.98}$ | $63.73_{\pm5.54}$ | $61.47_{\pm9.43}$ | $59.73_{\pm11.11}$ | $59.47_{\pm10.85}$ | $64.93_{\pm3.64}$ | $64.40_{\pm6.55}$ | $69.33_{\pm4.78}$ |
| **Recall** | $2.40_{\pm2.94}$ | $1.20_{\pm2.40}$ | $2.40_{\pm4.80}$ | $2.80_{\pm5.60}$ | $6.00_{\pm7.38}$ | $11.60_{\pm8.52}$ | $20.80_{\pm10.47}$ | $21.27_{\pm8.26}$ |

Table 3: Ablation study on different loss items for Deep Signature. We report accuracy and recall for performance evaluation.

| $\mathcal{L}_u$ | $\mathcal{L}_t$ | $\mathcal{L}_c$ | Accuracy | Recall |
|---|---|---|---|---|
| × | × | ✓ | $67.60_{\pm6.02}$ | $19.60_{\pm6.38}$ |
| × | ✓ | ✓ | $67.97_{\pm1.15}$ | $17.60_{\pm4.45}$ |
| ✓ | × | ✓ | $66.00_{\pm2.23}$ | $23.20_{\pm4.99}$ |
| ✓ | ✓ | ✓ | $69.33_{\pm4.78}$ | $21.27_{\pm8.26}$ |

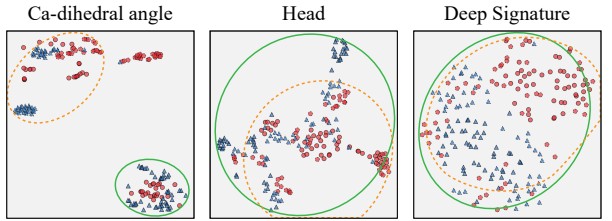

Figure 5: Visualization of the extracted features via t-SNE. Positive samples are presented in red circles and negative samples are in blue triangles.

by incorporating the log-signature transform into our method design, demonstrating the importance of capturing interatomic interaction dynamics for property prediction. Moreover, we validate the scalability of our approach to dynamical system across various time scales and different system sizes. As shown in Fig. 4(a), when extending the time duration of dynamics from 1s to 5s, our Deep Signature consistently obtains the best performance across these settings. Furthermore, when we change the system size via increasing the number of nodes that would enhance the complexity of dynamics, the results in Fig. 4(b) show that our method can still perform well under these conditions. These results showcase the good scalability of our method for dealing with dynamical systems of different timescales and sizes. In contrast, while GraphLSTM yields competitive results for systems with fewer than 250 nodes, a notable performance degradation appears as the number of nodes keeps increasing. We conjecture that this is because the averaged dynamics become less representative of the global dynamics as the system scale grows.

## 4.2 EXPERIMENTS ON EPIDERMAL GROWTH FACTOR RECEPTOR MUTATION DYNAMICS

In this study, we investigate the binding process between EGFR mutations and EGFR-receptor tyrosine kinases (RTK) during which their interactions can contribute to drug resistance mechanisms. The combinations between four RTK partners and five mutation types together with wild type as a reference are considered. To acquire the dynamic data, we follow the pipeline presented in (Zhu et al., 2021) that each system first undergoes energy minimization prior to simulation, then it is heated for 100 ps, followed by density equilibration for 100 ps and constant pressure equilibration for 5 ns, and finally runs simulation on the equilibrated structures for 50 ns. This results in 24 trajectories, each comprising 1,000 frames describing the temporal interactions among approximately 5,000 atoms. Each trajectory is labeled according to its sensitivity towards the drug. Here, our deep spectral clustering model consists of three layers that progressively coarsen the dynamics into 400, 200, 50 nodes. We set the scaling parameters as $\lambda_1 = 1$, $\lambda_2 = 0.01$, and $\lambda_3 = 10$. The model is trained for 200 epochs with an initial learning rate 5e-5 and a weight decay of 1e-5. In addition to the deep learning-based baselines introduced above, we further consider two baselines that extract dihedral angles as geometrical descriptors, followed by PCA to reduce the dimension of features and SVM for classification. The overall comparison results are summarized in Table 2.

As can be seen, our Deep Signature method consistently obtains the best classification results on EGFR dynamics, achieving an accuracy of 69.33% and a recall of 21.27%. This validates the practical applicability of our method for its capability to tackle complex dynamics inherent in large-scale molecular systems. In addition, in contrast to baseline methods that achieve high accuracy but low recall which often due to their reliance on bypasses caused by the varying sample sizes for different classes, our method demonstrates a substantial improvement in recall, indicating its robustness under class imbalance issues. Furthermore, when compared to conventional geometrical descriptor methods, deep learning-based approaches, with the exception of ours, tend to underperform on this dataset. This discrepancy can be attributed to the intricate non-linearity and atomic fluctuations

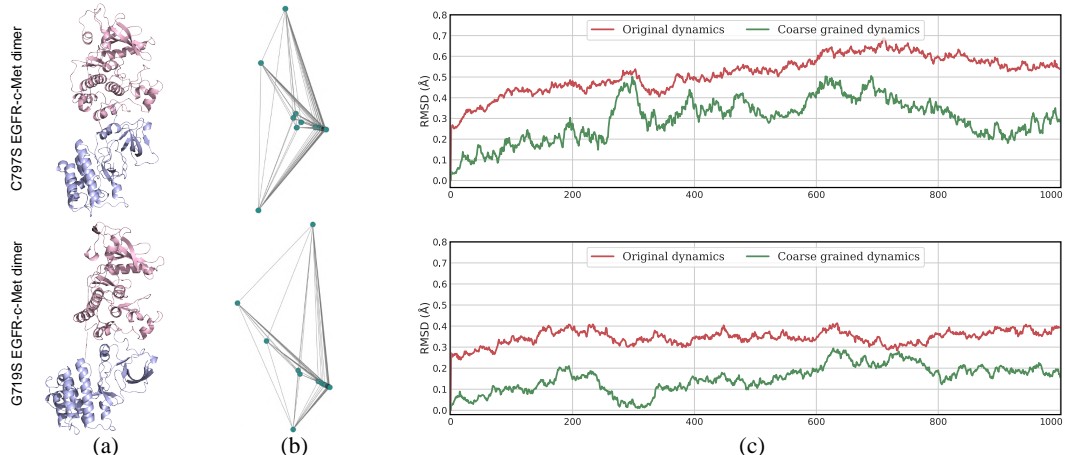

Figure 6: The effect of the deep spectral clustering module for C797S and G719S mutated EGFR dimers. (a) are origianl 3D conformations, (b) are coarse grained graphs, (c) are the RMSD curves of raw trajectories and coarse grained trajectories.

characteristic of EGFR dynamics, which pose significant challenges for deep learning techniques to extract meaningful patterns from a limited amount of data.

**Ablation study.** We conduct an ablation study to assess the necessity of each loss term used in our model, with the results for EGFR dynamics presented in Table 3. We begin by training our Deep Signature model using only the classification loss $\mathcal{L}_c$, which yields an accuracy of $67.60\%$. Upon incorporating the temporal consistency loss $\mathcal{L}_t$, the accuracy increases to $67.97\%$. However, when we replace $\mathcal{L}_t$ with the spectral clustering loss $\mathcal{L}_u$, performance slightly degrades to $66.00\%$. This decrease may be attributed to the tendency of $\mathcal{L}_u$ to guide the spectral clustering module towards hard assignments, thereby impairing the representation ability of the coarse grained dynamics. Training with all three loss terms produces the best results, demonstrating the validity of our method design.

**Analysis on deep spectral clustering module.** To understand the impact of our spectral clustering module on the graph structures and trajectories for EGFR dynamics, we investigate two cases: a drug-resistant C797S mutant and a drug-sensitive G719S mutant respectively dimerized with EGFR partners, with their results visualized in Fig. 6. By comparing (a) and (b), we observe that coarse grained graphs produced by our module can maintain the overall structure of original conformations, and the nodes mainly distribute in the middle region, which is valid since interatomic interactions that facilitate protein function commonly occur in this area. Besides, as shown in (c), the root mean square deviation (RMSD) curves for both the raw and coarse grained trajectories generally follow the same trend, indicating a high fidelity of the coarse grained dynamics to the original dynamics.

**Analysis on path signature transform module.** We visualize the feature space via t-SNE to explore the discriminability of learned features after path signature transform. The results of all data samples with colors indicating their classes are presented in Fig. 5. As seen, both the non-learnable features, which compute dihedral angles for all $C\alpha$ atoms, and the learned features derived solely from the first frame tend to gather tightly in feature space, making it difficult to establish a decision boundary to distinguish samples belonging to different classes. In contrast, features extracted via our Deep Signature method are more uniformly distributed, and we can find such a decision boundary easily, verifying the good discriminability of these features. We further use green circles to indicate training samples and orange dashed circles to denote test samples in Fig. 5. These two circles typically exhibit limited overlap due to distribution shift between the training and test samples, which arises from heterogeneity in the number of atoms and molecular topological structures. Nevertheless, Deep Signature exhibits an ability to learn generalizable features for unseen samples. This is because learning features that respect symmetry enhances generalization capability (Schütt et al., 2017b), and incorporating layer normalization helps mitigate the distribution discrepancy (Ba et al., 2016).

**Analysis on interpretability.** Our approach also exhibits notable interpretability for its ability to identify every possible type of interaction among atoms that are essential for protein functioning. This capability stems from the deep spectral clustering module which only involves linear mapping for cluster assignment, and log-signature features that maintain permutation equivariance as dis-

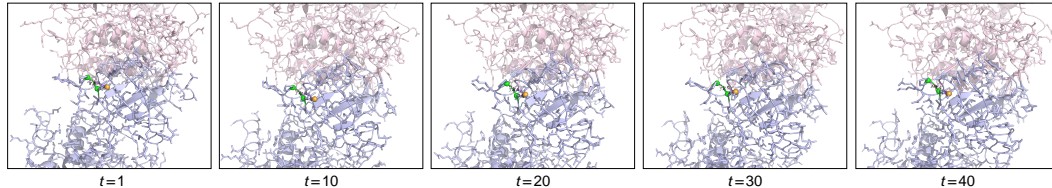

Figure 7: The visualization of critical pathways and interatomic interactions that contribute to drug sensitiveness on the EGFR dynamics.

Table 4: Comparison of different methods for the classification results on the GPCR dynamics.

| Method | Dihedral angle | Cα-dihedral angle | Head | Tail | Head & Tail | GraphLSTM | GraphTrans | Deep Signature |
|---|---|---|---|---|---|---|---|---|
| **Accuracy** | $52.67_{\pm0.94}$ | $44.00_{\pm1.12}$ | $53.07_{\pm6.47}$ | $47.20_{\pm5.86}$ | $47.20_{\pm5.86}$ | $53.47_{\pm14.86}$ | $44.80_{\pm6.97}$ | $58.00_{\pm4.17}$ |
| **Recall** | $31.73_{\pm1.77}$ | $35.47_{\pm3.73}$ | $26.40_{\pm13.21}$ | $37.60_{\pm15.72}$ | $37.60_{\pm15.72}$ | $38.13_{\pm13.19}$ | $33.33_{\pm21.08}$ | $43.30_{\pm7.85}$ |

cussed in Section 3.3. In practice, we apply the Gradient ⊙ Input method (Shrikumar et al., 2017) to quantify the contribution of each element in log-signature feature to the final prediction output and then identify three key atoms whose interactive dynamics play a pivotal role in the progression of EGFR. The identified atoms, along with their interatomic distances, are illustrated in Fig. 7. Notably, these identified atoms fall within the hinge region that comprises all ATP binding sites, demonstrating strong consistency with the experimental observation (Kaufman et al., 2021).

### 4.3 EXPERIMENTS ON G PROTEIN-COUPLED RECEPTORS DYNAMICS

The GPCR superfamily is a major therapeutic target as its functioning regulates nearly every physiological process in the human body. To create a dataset for its analysis, we select 26 structures with their MD simulations from the Molecular Dynamics Database for GPCRs (GPCRmd) (Rodríguez-Espigares et al., 2020), including 13 active state structures and 13 inactive state structures. The task is to identify these distinct active states. Each simulation is conducted for 500 ns with a time interval of 200 ps, therefore each trajectory consists of 2,500 frames that describe the conformational dynamics. Here, we only consider atoms in backbones. Experimental results are provided in Table 4.

It is evident that Deep Signature achieves the highest performance on the GPCR dynamics, with an accuracy of 58.00% and a recall ratio of 43.30%. The results are significantly better than the compared baselines. Traditional geometrical descriptor methods, such as the dihedral angles and Cα-dihedral angles, exhibit relatively lower classification accuracy and recall compared to our method. This again verifies that geometrical descriptors are inadequate for capturing dynamic patterns when the dynamics are highly nonlinear due to the existence of atomic fluctuations in GPCR dynamics. Moreover, relying solely on the first or last frame fails to account for the dynamic nature of interactions, thereby limiting the ability to discern useful patterns for predicting structural states. Although the GraphLSTM method outperforms the geometric descriptor methods, achieving a moderate accuracy of 53.47% and a recall ratio of 38.13%, it still falls short of the superior classification capabilities exhibited by the Deep Signature method, demonstrating the effectiveness of our approach in characterizing the complex dynamics of GPCR systems.

## 5 CONCLUSION

In this paper, we introduce a novel deep learning framework, Deep Signature, to deal with the long-standing challenge of understanding protein dynamics in large-scale biological systems. It comprises soft spectral clustering to aggregate cooperative dynamics and log-signature transformation to characterize global interactive dynamics. Theoretically, our method exhibits desirable properties such as invariance to translation, near-invariance to rotation, and equivariance to atomic coordinate permutation. Experimental results on three biological process benchmarks verify the effectiveness of Deep Signature in capturing the complex interactive dynamics of large-scale molecular systems. We hope our work can offer a promising new direction for the analysis of protein dynamics.

## ACKNOWLEDGMENTS

This work was supported in part by the Hong Kong Innovation and Technology Commission (ITC) (InnoHK Project CIMDA), in part by the Institute of Digital Medicine of City University of Hong Kong (Project 9229503), in part by the Hong Kong Research Grants Council under Projects 21200522, 11200323 and 11203220, in part by Chow Sang Sang Donation and Matching Fund (Project 9229161), and in part by the Hong Kong Innovation and Technology Commission (Project GHP/044/21SZ).

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

# Table of Contents

## A    PROOF FOR PROPERTIES

In the beginning of our proof, we would first decompose our feature extracting process into two operations $g_{\text{GNN}}(\cdot)$ and $\text{LogSig}(\cdot)$ as presented in Section 3.2 such that the overall feature transform is their composition $g_{\text{GNN}} \circ \text{LogSig}$. This decomposition offers us the opportunity to analyze the properties of component separately. Besides, since the logarithm map is bijective, which implies there is one-to-one correspondence between the signature and the log-signature (Lyons et al., 2007), thus we can turn to analyze signature transform for intuitive demonstration.

### A.1    TRANSLATION INVARIANCE

Given trajectory data $\mathbf{X}_{1:T} \in \mathbb{R}^{T \times N \times 3}$, let $\mathcal{T}_B$ represent a translation matrix $B \in \mathbb{R}^{N \times 3}$ on the trajectory data such that at a certain time stamp $t$ we have $\mathcal{T}_B(\mathbf{X}_t) = \mathbf{X}_t + B$. For the coarse grained dynamics acquired by $\tilde{\mathbf{X}}_t^{\text{pool}} = \mathbf{Q}^T \mathbf{X}_t$ as presented by Eq. (5), as $\mathbf{Q}^T$ is a linear matrix, we can get that $\mathbf{Q}^T \mathcal{T}_B(\mathbf{X}_t) = \mathbf{Q}^T \mathbf{X}_t + \mathbf{Q}^T B = \mathcal{T}_{\mathbf{Q}^T B}(\mathbf{Q}^T \mathbf{X}_t)$, thus $\tilde{\mathbf{X}}_{1:T}$ maintain equivariance with respect to translation $\mathcal{T}_B$ on input trajectory $\mathbf{X}_{1:T}$. Besides, as presented in Eq. (8) that the path signature is composed of iterated path integrals, it inherits the properties of translation invariance that $\text{Sig}_{a,b}^D(\tilde{\mathbf{X}} + c) = \text{Sig}_{a,b}^D(\tilde{\mathbf{X}})$. In the whole, our path signature features $\text{Sig}_{a,b}^D(\tilde{\mathbf{X}})$ are translation-invariant with respect to input trajectory data $\mathbf{X}_{a,b}$.

### A.2    ROTATION INVARIANCE

Given the coarse grained dynamics acquired by $\tilde{\mathbf{X}}_t = \mathbf{Q}^T \mathbf{X}_t$, where $\mathbf{X}_t \in \mathbb{R}^{N \times 3}$ and $\tilde{\mathbf{X}}_t \in \mathbb{R}^{M \times 3}$. The cluster assignment matrix $\mathbf{Q} \in \mathbb{R}^{N \times M}$ holds that $\mathbf{Q}\mathbf{1}_M = \mathbf{1}_N$ and $\mathbf{Q}^T \mathbf{Q} = \mathbf{I}_M$. For any orthogonal matrix $\mathbf{R}_\theta \in \mathbb{R}^{3 \times 3}$ performed on $\mathbf{X}_t$ termed as $\mathbf{R}_\theta \mathbf{X}_t = (\mathbf{R}_\theta \mathbf{X}_t^1, \dots, \mathbf{R}_\theta \mathbf{X}_t^N)$, we proof below that rotating the input results in an equivalent rotation of the output $\mathbf{R}_\phi \tilde{\mathbf{X}}_t = \mathbf{Q}^T \mathbf{R}_\theta \mathbf{X}_t$, where $\mathbf{R}_\phi \in \mathbb{R}^{M \times 3}$ is an orthogonal matrix performed on $\tilde{\mathbf{X}}_t$. Let we start with the right-hand side $\mathbf{Q}^T \mathbf{R}_\theta \mathbf{X}_t$. Here, $\mathbf{Q}^T$ has the form $\mathbf{Q}^T = (\mathbf{Q}_1^T, \dots, \mathbf{Q}_M^T)$, where the assignment function for $m$-th cluster is $\mathbf{Q}_m^T \in \mathbb{R}^N$, $m \in \{1, \dots, M\}$, and the corresponding aggregated coordinate is $\mathbf{Q}_m^T \mathbf{X}_t$ which is a linear combination in each dimension. Due to the fact that $\mathbf{Q}_m^T \mathbf{R}_\theta \mathbf{X}_t$ can always be represented by $\mathbf{R}_\phi \mathbf{Q}_m^T \mathbf{X}_t$, which implies that for a rotation transform $\mathbf{R}_\theta$ on the original conformational space, there always exists an equivalent rotation transformation $\mathbf{R}_\phi$ in coarse grained conformational space. We then accomplish our proof that $\mathbf{R}_\phi \tilde{\mathbf{X}}_t = \mathbf{Q}^T \mathbf{R}_\theta \mathbf{X}_t$.

After that, we turn to analyze the effect of rotation on features acquired by signature transform. Unfortunately, not all elements of a path signature feature exhibit rotation invariance. As demonstrated by Diehl (2013), rotation invariants only exist on levels of even order. For a 2-dimensional continuous path $X : [a, b] \to \mathbb{R}^2$ as presented in Fig. 8, the depth-1 terms corresponding to the variations for each channel over the interval are represented by $\Delta X_1$ and $\Delta X_2$, while the depth-2 term corresponding to the signed area between the chord connecting the endpoints and the real path is denoted as $A$. The calculation of depth-2 term corresponds to the coefficient of the polynomial $[e_n, e_m]$ in Eq. (10)

$$A = A_+ - A_- = \frac{1}{2}\left(S_{a,b}^{1,2}(X) - S_{a,b}^{2,1}(X)\right).$$

Here, the sign of area exactly indicates the orientation of acceleration, while an increase in the absolute value of surrounding accelerations would increase the proportion of the represented area accordingly. Obliviously, the area $A$ is rotation invariant, while $\Delta X_1, \Delta X_2$ do not hold this property. As suggested by Diehl (2013), a depth-2 term in Eq. (10) is rotation invariant only if it involves the iterated integrals over different channels. Although further investigation into rotation invariants

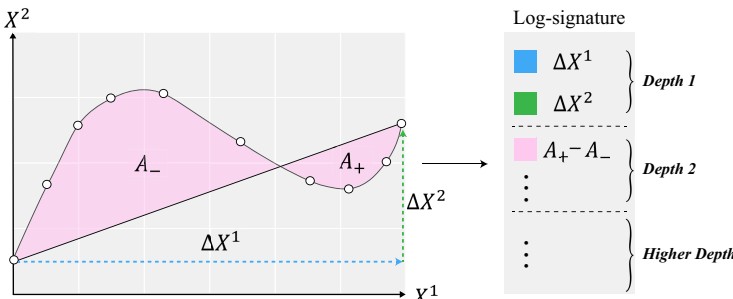

Figure 8: Geometric intuition for the first two levels of the log-signature on a 2-dimensional path. We can observe that the depth-1 terms represent the change in each of the coordinates over the interval, and the depth-2 term corresponds to the *Lévy area* of the path, shown as the signed area enclosed by the curve and the chord connecting its start and endpoints.

for path signature with higher depth over beyond two dimensional paths is still lacking, current conclusion on rotation invariance acquired for depth-2 log-signauture can readily extend to higher dimensional paths.

In our experimental setup, given the coarse grained dynamics $\tilde{\mathbf{X}}_{a,b} \in \mathbb{R}^{T \times 3M}$, the dimension of depth-$D$ log-signature is

$$\dim(\text{LogSig}_{a,b}^D(\tilde{\mathbf{X}})) = \sum_{d=1}^{D} \frac{1}{d} \sum_{i|d} \mu\left(\frac{d}{i}\right)(3M)^i,$$

where $\mu$ is the Möbius function defined as

$$\mu(n) = \begin{cases} 0 & \text{if } n \text{ has one or more repeated prime factors} \\ 1 & \text{if } n = 1 \\ (-1)^k & \text{if } n \text{ is the product of } k \text{ distinct prime numbers} \end{cases}$$

When we specify $d = 2$, the ratio of rotation-invariant elements over the whole log-signature feature can then be calculated as

$$\gamma = \frac{3M - 1}{3M + 1},$$

which indicates the majority of features are rotation-invariant for a large-scale molecular system as we focus on in this paper.

## A.3 PERMUTATION EQUIVARIANCE

For the deep clustering module implemented by GNNs $g_{\text{GNN}}(\cdot)$ with the formula presented in Eq. (1) as follows

$$\mathbf{H}^l = \sigma(\tilde{\mathbf{D}}^{-1/2}\tilde{\mathbf{A}}\tilde{\mathbf{D}}^{-1/2}\mathbf{H}^{l-1}\mathbf{W}_{\text{GNN}}^{l-1}).$$

For one layer, the calculation of node embeddings per node involves sum over contributions from different atoms. As a result, node embedding matrix $\mathbf{H}^l$ naturally exhibit equivariance with respect to the permutation symmetries of graphs. In addition, the linear mapping shown in Eq. (5) only works for dimension alignment, making no influence on the permutation equivariance. So the coarse grained dynamics $\tilde{\mathbf{X}}_{1:T}$ is permutation equivariant with respect to an input set of atoms.

We then delve into log-signature transform by examining Eq. (10). Notably, the monomials such as $e_n$ and $[e_n, e_m]$ constitute the basis vectors of the vector space, and their coefficients are calculated using iterate integrals over paths indexed by these monomials. Ideally speaking, log-signature features hold permutation invariance since we can arbitrarily select and order the monomials. Such phenomenon becomes even clearer when we revisit the signature transform in Eq. (7). In the signature transform, the superscripts indicating the paths traverse the set of all multi-indexes denoted as

$$W = \{(j_1, \ldots, j_d) | d \geq 1, j_1, \ldots, j_d \in \{1, \ldots, 3M\}\}.$$

Hence, the order of the multi-indexes has no impact on the captured dynamic information. In practical implementations, we organize the multi-indexes in ascending order to collect these iterated

integral terms (Kidger & Lyons, 2021). Consequently, a permutation on the indices of input atoms would yield a predictable permutation transform on the indices of log-signature terms. As a result, we drive the conclusion that our Deep signature method exhibits permutation equivariance with respect to input atom indices.

## A.4 TIME-REPARAMETRIZATION INVARIANCE

Since coarse graining will maintain temporal consistency, we only need to analyze the invariance under time-reparametrization for coarse grained dynamics. A reparametrization of a coarse grained pathway $\mathbf{X} : [a, b] \to \mathbb{R}^{3M}$ is a path $\check{\mathbf{X}} : [a, b] \to \mathbb{R}^{3M}$ where $\check{\mathbf{X}}_t = \mathbf{X}_{\psi_t}$ where $\psi$ is a subjective, continuous, non-decreasing function $\psi : [a, b] \to [a, b]$. We have the following theorem

**Theorem 1** *Let* $\check{\mathbf{X}}_t$ *be a reparametrization of* $\mathbf{X}$*, then we have* $\mathrm{Sig}(\check{\mathbf{X}}) = \mathrm{Sig}(\mathbf{X})$*.*

Consider two real-valued paths $X, Y : [a, b] \to \mathbb{R}$ and a surjective, continuous, non-decreasing reparametrization function over time $\psi : [a, b] \to [a, b]$. Then we have reparametrized paths $\check{X}, \check{Y} : [a, b] \to \mathbb{R}$ by $\check{X}_t = X_{\psi_t}$ and $\check{Y}_t = Y_{\psi_t}$

$$\int_1^T \check{Y}_t d\check{X}_t = \int_1^T Y_{\psi_t} dX_{\psi_t} = \int_1^T Y_{\psi_t} \dot{X}_{\psi_t} \dot{\psi}_t dt = \int_1^T Y_{\psi_t} \dot{X}_{\psi_t} d\psi_t$$

After replacing $u = \psi_t$, we have $\int_1^T \check{Y}_t d\check{X}_t = \int_1^T Y_u dX_u$, which means path integrals are invariant under a time reparametrization of both paths.

Since every term of the signature $S(\mathbf{X})_{a,b}^{j_1,\dots,j_d}$ is defined as an iterated path integral of $\mathbf{X}$, it follows from the above that

$$S(\check{\mathbf{X}})_{a,b}^{j_1,\dots,j_d} = S(\mathbf{X})_{a,b}^{j_1,\dots,j_d}, \quad \forall k \geq 0, \ j_1, \dots, j_d \in \{1, \dots, 3M\}$$

This complements the proof.

# B EXPERIMENTAL SETTINGS

## B.1 GENE REGULATORY DYNAMICS

The dynamics for gene regulatory networks are governed by Michaelis-Menten equation as follows,

$$\frac{\mathrm{d}\boldsymbol{x}_t(v_i)}{\mathrm{d}t} = -b_i \boldsymbol{x}(v_i)^f + \sum_{j=1}^n \mathbf{A}^{(i,j)} \frac{\boldsymbol{x}^h(v_j)}{\boldsymbol{x}^h(v_j) + 1}, \tag{13}$$

where the first term models the degradation when $f = 1$ or dimerization when $f = 2$, and the second term represents genetic activation, with the Hill coefficient $h$ determining the level of cooperation in the regulation of the gene.

**Data generation.** For the gene regulatory dynamics, we curate a dataset consisting of 100 trajectories that delineate the intricate interactive dynamics between genes and transcription factors. These trajectories are divided into two distinct categories: degradation type ($f = 1$) and dimerization type ($f = 2$), each encompassing an equal number of trajectories. To commence a simulation, we first initialize a graph network featuring 100 nodes using a Power-law network generator to elucidate the structural interconnections between these nodes. Subsequently, we employ the Dormand-Prince method to numerically solve the gene regulatory system described by Eq. (13), with a simulation duration of 2 seconds and a time interval of 0.004 seconds. This computation simulation yields trajectories comprising 500 frames each, laying the foundation for our subsequent experiments.

**Model architecture.** We implement the deep spectral clustering module in our approach using two graph pooling layers, which first coarsen the dynamics into 60 nodes and subsequently into 30 nodes. The dimension of hidden states for the stacked GCN layers is kept as 10. For a tiny implementation, we simply extract the log-signature features for the coarse grained dynamics and then employ a two-layer MLP for property prediction.

**Training details.** We train our model with mini-batches of size 48 for 100 epochs using Adam with the initial learning rate of 5e-4 and a weight decay 1e-4. The coefficients of loss terms are set as $\lambda_1 = 1$, $\lambda_2 = 0.01$, and $\lambda_3 = 10$.

## B.2 EPIDERMAL GROWTH FACTOR RECEPTOR MUTATION DYNAMICS

**Data generation.** In this investigation, we delve into the intricate binding dynamics between EGFR mutations and RTK, exploring how their interactions can trigger mechanisms of drug resistance. The study encompasses the amalgamation of four RTK partners and five distinct mutation types, alongside the wild type serving as a reference point. To capture the dynamic essence of these interactions, we adhere to the methodological framework outlined in (Zhu et al., 2021), wherein each system undergoes a meticulous pre-simulation process. This involves an initial energy minimization phase, followed by a 100 picosecond heating stage, subsequent density equilibration spanning 100 picoseconds, and a further 5 nanoseconds of constant pressure equilibration. The equilibrated structures then undergo a simulation period of 50 nanoseconds, resulting in the generation of 24 trajectories. Each trajectory comprises 1000 frames, detailing the temporal interplay among approximately 5,000 atoms. Notably, these trajectories are categorized based on their sensitivity or resistance to the administered drug. For the MD simulations, we leverage the explicit-solvent model integrated within the Amber software suite, utilizing the *Ff99SB* and *gaff* force fields to drive the simulations.

**Model architecture.** For the deep spectral clustering module, we utilize three graph pooling layers that progressively coarsen the dynamics into 400, 200, 50 nodes. The dimension of hidden states for the stacked GCN layers is kept as 20. For the path signature transform module, we partition the input coarse grained dynamics into four segments for the subsequent application of the log-signature transform.

**Training details.** The model is trained with mini-batches of size 16 for a total of 200 epochs. We employ the Adam optimizer with an initial learning rate of 5e-5 and a weight decay 1e-5 to optimize the model parameters. Additionally, the scaling parameters are specified as $\lambda_1 = 1$, $\lambda_2 = 0.01$, and $\lambda_3 = 10$.

## B.3 G PROTEIN-COUPLED RECEPTORS DYNAMICS

**Data generation.** We download our data from the GPCRmd (http://gpcrmd.org/) (Rodríguez-Espigares et al., 2020) database, which is an online platform that incorporates web-based visualization capabilities and shares data. This database includes at least one representative structure from each of the four structurally characterized GPCR classes, and holds more than 600 GPCR MD simulations from GPCRmd community and individual contributions. To create our dataset, we select 26 trajectories of the $\beta 2$AR-rh1 GPCR inactive (2RH1) and active (3P0G) receptor state with a full agonist. The receptor consists of 282 and 285 amino acids for inactive and active state respectively. Each simulation is conducted for 500 ns with a time interval of 200 ps, therefore every trajectory consists of 2,500 frames that describe the atomic 3D positions over time.

**Model architecture.** We implement the deep spectral clustering module in our approach using two graph pooling layers, which first coarsen the dynamics into 100 nodes and subsequently into 50 nodes. The dimension of hidden states for the stacked GCN layers is kept as 20. In the path signature transform module, the input coarse grained dynamics are partitioned into four segments to facilitate the subsequent application of the log-signature transform.

**Training details.** The model is trained with mini-batches of size 16 for a total of 200 epochs. We employ the Adam optimizer with an initial learning rate of 5e-5 and a weight decay 1e-5 to optimize the model parameters. Furthermore, the scaling parameters are specified as $\lambda_1 = 1$, $\lambda_2 = 0.01$, and $\lambda_3 = 10$.

# C  Further Results and Analysis

## C.1  Parameter Sensitivity Analysis

**The sensitivity of coarsening level.** To further validate the robustness of our method for different coarsening levels, we conducted additional experiments by varying the coarsening levels for the EGFR and GPCR datasets. The results are summarized in Table 5.

Table 5: Comparison of EGFR and GPCR Accuracy at Different Coarsening Levels

| Coarsening Level | 100 | 50 | 30 | 10 | 5 |
|---|---|---|---|---|---|
| **EGFR Accuracy (%)** | $66.93_{\pm4.62}$ | $69.33_{\pm4.78}$ | $66.40_{\pm5.06}$ | $63.20_{\pm7.56}$ | $57.33_{\pm4.73}$ |
| **GPCR Accuracy (%)** | $50.67_{\pm5.58}$ | $58.00_{\pm4.17}$ | $58.00_{\pm6.01}$ | $57.73_{\pm8.92}$ | $55.07_{\pm6.32}$ |

From these results, we observe that coarsening levels of 30 and 50 beads yield the best accuracy for both datasets, demonstrating the robustness of our method across different levels of coarsening. While extreme levels, such as 100 or 5 beads, result in reduced performance, the method is flexible and performs consistently well within a broad range of coarsening levels.

**The value of $\lambda_1$, $\lambda_2$ and $\lambda_3$.** In the objective of Deep Signature presented in Eq. (12), there exist three hyper-parameters $\lambda_1$, $\lambda_2$ and $\lambda_3$ reweighting the loss terms. We now conduct the parameter sensitivity test by varying their value in $\{10.0, 1.0, 0.1, 0.01\}$. The results are summarized in Table 6, 7 and 8, respectively.

Table 6: Sensitivity analysis for $\lambda_1$ and its effect on accuracy.

| $\lambda_1$ | 10.0 | 1.0 | 0.1 | 0.01 |
|---|---|---|---|---|
| **Accuracy (%)** | $59.60_{\pm8.34}$ | $69.33_{\pm4.78}$ | $67.87_{\pm1.95}$ | $67.33_{\pm2.77}$ |

Table 7: Sensitivity analysis for $\lambda_2$ and its effect on accuracy.

| $\lambda_2$ | 10.0 | 1.0 | 0.1 | 0.01 |
|---|---|---|---|---|
| **Accuracy (%)** | $65.27_{\pm2.48}$ | $67.67_{\pm1.63}$ | $67.80_{\pm1.15}$ | $69.33_{\pm4.78}$ |

Table 8: Sensitivity analysis for $\lambda_3$ and its effect on accuracy.

| $\lambda_3$ | 10.0 | 1.0 | 0.1 | 0.01 |
|---|---|---|---|---|
| **Accuracy (%)** | $69.33_{\pm4.78}$ | $66.53_{\pm2.65}$ | $62.00_{\pm4.22}$ | $58.93_{\pm1.71}$ |

As shown, our model is generally robust to moderate variations in the hyperparameters. For instance, $\lambda_1 = 1.0$, $\lambda_2 = 0.01$, and $\lambda_3 = 10.0$ consistently deliver strong performance. However, extreme values for these parameters (*e.g.*, $\lambda_3 = 0.01$ or $\lambda_1 = 10.0$) can lead to a noticeable drop in accuracy, highlighting the importance of choosing balanced values.

