# OpenReview forum: "Deep Signature: Characterization of Large-Scale Molecular Dynamics"
_ICLR.cc/2025/Conference — ICLR 2025 Poster_

### Official Review · Reviewer_xZWn · 2024-10-28

**Soundness:** 2
**Presentation:** 3
**Contribution:** 3
**Rating:** 6
**Confidence:** 2

**Summary:**

The paper presents a framework, Deep Signature, designed to analyze the dynamics and interatomic interactions of large-scale molecular system. The method uses a deep spectral clustering model to capture coarse grained dynamics, a path signature transform module to characterize interatomic interactions through iterated integrals, and a classifier for property prediction. Theoretically, Deep Signature is shown to maintain desirable symmetry properties. The method demonstrates improved accuracy across three benchmarks and demonstrates interpretability.

**Strengths:**

1. Authors develop an end-to-end framework to characterize interatomic interactions and dynamics of large-scale molecules, it shows improvement on three benchmarks and provides interpretability.
2. The size of the system is reduced by deep spectral clustering module without any expert knowledge.
3. The framework's desirable properties are supported by theoretical analysis.

**Weaknesses:**

1. The authors compare their approach to baseline methods, but a more comprehensive comparison with some SOTA baselines would provide a more robust evaluation.
2. The manuscript would benefit from a comparison of deep spectral clustering module with existing coarse graining methods.
3. An analysis of the model’s sensitivity to hyperparameters would provide insights into its robustness and reproducibility.

**Questions:**

1. How were the hyperparameters chosen, such as the loss coefficient parameters $\lambda_i$,  the number of nodes in deep spectral clustering model, and time interval $[r_i, r_{i+1}]$ in path signature transform?
2. When visualizing critical pathways and interatomic interactions on the EGFR dynamics, why were three atoms identified specifically? Could more than three be selected?
3. Have the authors considered comparing their approach with advanced time series classification algorithms？
4. How does the computational efficiency of the proposed method compare to baseline methods?

---

> ### Author Response · Authors · 2024-11-22
> **Thank you; Address your concerns [1/2]**
>
> Dear Reviewer,
>
> Thank you for dedicating your time and effort to reviewing our paper.
>
> ---
>
> **W1:** The authors compare their approach to baseline methods, but a more comprehensive comparison with some SOTA baselines would provide a more robust evaluation.
>
> **Response:** Thank you for your constructive comment. In this work, we approach a new learning problem of characterizing the dynamic information from atomic evolving trajectories for large molecules. Given the high complexity of nonlinear interactive dynamics across large spatial and temporal scales, and the non-ignorable requirements like physical symmetry, generalization towards heterogeneous structures, and interpretability, there exists no approach fully satisfying all these requirements, therefore we need to develop a tailored framework for large-scale MD data analysis.
>
> **W2:** The manuscript would benefit from a comparison of deep spectral clustering module with existing coarse graining methods.
>
> **Response:** Thank you for your valuable suggestion. Our proposed Deep Signature framework is highly flexible and allows for the replacement of the deep spectral clustering module with other advanced learning-based coarse-graining methods. However, what sets our deep spectral clustering module apart is its ability to dynamically aggregate local atomic interactions based on spectral graph theory, ensuring scalability to large systems while maintaining interpretability.
>
> To address your concern, we have conducted additional experiments on the GPCR dataset to compare our deep spectral clustering module with a commonly used radial ball approach for coarse-graining. The results are summarized below:
> | Method| Radial Ball (0.5A) | Radial Ball (1.0A) | Radial Ball (1.5A) | Deep Spectral Clustering |
> |---|:---:|:---:|:---:|:---:|
> | Number of Beads| ~150| ~50 | ~20 | 50|
> | Accuracy (%)| 56.40 ± 3.81 | 53.60 ± 2.37| 45.07 ± 3.34| 58.00 ± 4.17 |
>
>
> These results highlight that while alternative methods like the radial ball approach may effectively reduce the system's complexity, their inability to maintain a consistent representation across molecules limits their utility in downstream tasks. By contrast, our deep spectral clustering module not only ensures consistency but also optimally integrates with the surrogate task to enhance the overall performance of Deep Signature.
>
>
> **W3:** An analysis of the model’s sensitivity to hyperparameters would provide insights into its robustness and reproducibility.
>
> **Response:** Thank you for the valuable suggestion. To evaluate the robustness and reproducibility of our method, we conducted a series of experiments to analyze its sensitivity to hyperparameter selection. The results for the sensitivity analysis of the loss coefficient parameters ($\lambda_1$, $\lambda_2$, and $\lambda_3$) are provided below:
> | $\lambda_1$          | **10.0**       | **1.0**        | **0.1**        | **0.01**       |
> |:----------------:|:--------------:|:--------------:|:--------------:|:--------------:|
> | Accuracy (%) | 59.60 ± 8.34   | 69.33 ± 4.78   | 67.87 ± 1.95   | 67.33 ± 2.77   |
>
>
> | $\lambda_2$          | **10.0**       | **1.0**        | **0.1**        | **0.01**       |
> |:----------------:|:--------------:|:--------------:|:--------------:|:--------------:|
> | Accuracy (%)| 65.27 ± 2.48   | 67.67 ± 1.63   | 67.80 ± 1.15   | 69.33 ± 4.78   |
>
>
> | $\lambda_3$|10.0 |1.0|0.1|0.01|
> |:----------------:|:--------------:|:--------------:|:--------------:|:--------------:|
> | Accuracy (%)| 69.33 ± 4.78   | 66.53 ± 2.65   | 62.00 ± 4.22   | 58.93 ± 1.71   |
>
> In addition, we analyzed the effect of coarsening levels by varying the number of CG beads. The results are as follows:
> | Coarsening Level |100|50|30| 10|5|
> |---|---|---|---|---|---|
> | EGFR Accuracy (%) |66.93±4.62| 69.33 ± 4.78| 66.40 ± 5.06| 63.20 ± 7.56| 57.33 ± 4.73|
> | GPCR Accuracy (%) |50.67±5.58| 58.00 ± 4.17| 58.00 ± 6.01| 57.73 ± 8.92| 55.07 ± 6.32|
>
> As we can see, both 30 and 50 can lead to desired results, showing the robustness of our method across different levels of coarsening. We have incorporated these discussions into the revised version of the manuscript. Thank you again for your suggestion, which has helped improve the clarity and depth of our work.

---

> ### Author Response · Authors · 2024-11-22
> **Thank you; Address your concerns [2/2]**
>
> **Q1:** How were the hyperparameters chosen, such as the loss coefficient parameters λi, the number of nodes in deep spectral clustering model, and time interval [ri,ri+1] in path signature transform?
>
> **Response:** Thank you for the question. For all experiments, we set the loss coefficient parameters as $\lambda_1 = 1.0$, $\lambda_2 = 0.01$, and $\lambda_3 = 10.0$. The number of CG beads was fixed as 50 for both the EGFR and GPCR datasets. The time interval $[r_{i},r_{i+1}]$ in the path signature transform was set to 1 fs, aligning with the setup used for performing MD simulations. We presented these training details in our Appendix to save space.
>
> **Q2:** When visualizing critical pathways and interatomic interactions on the EGFR dynamics, why were three atoms identified specifically? Could more than three be selected?
>
> **Response:** Good question! In Fig. 7, we selected three atoms that contribute the most to the final prediction to enhance visualization clarity and focus on the most critical pathways and interatomic interactions. However, our approach is not restricted to three atoms. We can rank all interatomic interactions descendly based on their contributions to the prediction.
>
> **Q3:** Have the authors considered comparing their approach with advanced time series classification algorithms？
>
> **Response:** Thank you for your valuable question. In this work, we approach a new learning problem of characterizing the dynamic information from atomic evolving trajectories for large molecules. Given the high complexity of nonlinear interactive dynamics across large spatial and temporal scales, and the non-ignorable requirements like physical symmetry, generalization towards heterogeneous structures, and interpretability, existing time series classification algorithms cannot fully satisfying all these requirements, therefore we need to develop a tailored framework for large-scale MD data analysis.
>
> **Q4:** How does the computational efficiency of the proposed method compare to baseline methods?
>
> **Response:** Thank you for your question. To evaluate the computational efficiency of our method, we have conducted additional experiments comparing the inference time of baseline methods with our Deep Signature framework on the EGFR dataset. The results are summarized below:
> | Method |Head|Head & Tail|GraphLSTM| Deep Signature |
> |:---:|:---:|:---:|:---:|:---:|
> | Inference Time (s) | 4.97| 2.03| 2.09| 3.44 |
>
> As shown in the table, the inference time of Deep Signature is competitive with the baseline methods, demonstrating the efficiency of our method in processing large-scale molecular dynamics while maintaining strong predictive performance.

---

> ### Author Response · Authors · 2024-11-25
> **Thank you for reviewing**
>
> Dear Reviewer,
>
> Thank you for dedicating your time and effort to reviewing our paper again. We have carefully addressed your valuable comments in detail. As the public discussion phase is coming to an end soon, we kindly invite any further comments or suggestions you may have.
>
> We sincerely appreciate your efforts, which have significantly contributed to improving our manuscript.
>
> Kind regards,
> The Authors

---

> > ### Comment · Reviewer_xZWn · 2024-11-25
> >
> > Thank you for your detailed response. Most of my concerns have been addressed. I have decided to raise my score.

---

> > > ### Author Response · Authors · 2024-11-26
> > > **Thank you so much**
> > >
> > > Dear reviewer,
> > >
> > > We are glad to hear that our response helps address your concern. Your comment has greatly improved the quality of our paper. Thank you for your time and effort to review our work!

---

### Official Review · Reviewer_9koh · 2024-11-02

**Soundness:** 2
**Presentation:** 3
**Contribution:** 2
**Rating:** 6
**Confidence:** 3

**Summary:**

In the paper, the authors introduce the Deep Signature framework to capture complex dynamics using the evolution trajectories. The Deep Signature framework includes spectral clustering, signature transform and a classifier. Additionally, the authors show that the framework satisfies the desired symmetry constraints. Experiments on  gene regulatory dynamics, EGFR mutation dynamics and GPCR dynamics exhibit the empirical performance of the framework.

**Strengths:**

1. The presentation of the method is very clear and easy to understand.
2. The motivation of applying signature transform is reasonable.

**Weaknesses:**

1. Lack of experiments on large dataset. As the paper claims, the Deep Signature framework can capture large-scale complex dynamics. So I think experiments on datasets with large amount of data and system size are necessary. But the paper only includes the experiments on datasets with large system size.

2. I think the baseline in this paper is too weak. For example, the author should compare the strong baseline with graph transformer architecture[1] based on the first/last frame of the trajectory. Comparations between these strong baseline may strengthen the empirical performance of the framework.

[1].Ying, Chengxuan, et al. "Do transformers really perform badly for graph representation?." Advances in neural information processing systems 34 (2021): 28877-28888.

**Questions:**

1. I think the classification task using evolution trajectories is not a common setting. Could you explain why this problem setting is reasonable comparing to the one frame classification setting?  Do we really need the complete trajectory to do classification?

2. I do not understand your setting in the EGFR dynamics experiment since I'm not an expert on this domain. Could you please explain why the trajectory can be labeled according to its sensitivity towards the drug? I think the sensitivity should be a number rather than a 0/1 label.

---

> ### Author Response · Authors · 2024-11-22
> **Thank you; Address your concerns**
>
> Dear Reviewer,
>
>
> Thank you for dedicating your time and effort to reviewing our paper.
>
> ---
>
> **W1:** Lack of experiments on large dataset. As the paper claims, the Deep Signature framework can capture large-scale complex dynamics. So I think experiments on datasets with large amount of data and system size are necessary. But the paper only includes the experiments on datasets with large system size.
>
> **Response:** We appreciate the reviewer's suggestion to include experiments on a larger dataset, as this would indeed strengthen our study. However, to the best of our knowledge, there exist no other MD dataset larger than EGFR and GPCR in the biomolecular community. The limited availability of such datasets arises from the difficulty in collecting MD simulation data that meet specific biological criteria. Take the EGFR dataset as an example, each simulation trajectory corresponds to a unique mutation-induced conformational change in an EGFR dimer transitioning from non-equilibrium to equilibrium states. Some dynamic processes, such as binding pose and hydrogen bond variations, are essential for deciphering the drug resistance mechanism in lung cancer therapy. However, the dataset is constrained by the fact that only approximately 100 mutations have been identified, and the drug sensitivity of some of these mutations remains to be experimentally validated. We acknowledge the importance of validating our results on larger datasets and plan to enlarge the EGDR dataset in future research.
>
>
> **W2:** I think the baseline in this paper is too weak. For example, the author should compare the strong baseline with graph transformer architecture[1] based on the first/last frame of the trajectory. Comparations between these strong baseline may strengthen the empirical performance of the framework.
>
> **Response:** Thanks for the valuable suggestion. This work emphasizes on the extraction of dynamic information to decipher protein function, which is conceptually parallel to molecular property prediction approaches based on static single frame. To further address your concern, we have conducted additional experiments using the recommended Graphormer model on the GPCR dataset. Our implementation is based on https://github.com/leffff/graphormer-pyg, and it cannot be incorporated with our deep spectral clustering module to reduce the scale of the molecule as it requires a sparse graph input. The results are summarized below:
> | Model          | Accuracy (%)      | Recall (%)       | Total Inference Time (s) |
> |---|---|---|:---:|
> | Graphormer     | 43.33 ± 6.12      | 39.60 ± 15.85 |  282.12  |
> | Deep Signature | 58.00 ± 4.17     | 43.30 ± 7.85  |     0.31   |
>
> As shown, Graphormer struggles to converge effectively due to the difficulty in extracting meaningful patterns from large-scale molecular systems. Besides, its inference speed is significantly hindered by the computational complexity of shortest-path calculations. These results highlight the necessity of developing tailored frameworks specifically designed for large-scale molecular systems to address these challenges effectively.
>
> **Q1:** I think the classification task using evolution trajectories is not a common setting. Could you explain why this problem setting is reasonable comparing to the one frame classification setting? Do we really need the complete trajectory to do classification?
>
> **Response:** Thank you for your question. Deciphering the protein function mechanisms from molecular evolving trajectories is a longstanding and fundamental challenge in physical biology. Previous research has identified that dynamic information, such as protein side-chain motions, binding interactions, conformational transition pathways, and hydrogen bond vibrations, is critical for protein function [1, 2, 3]. In this work, we are the first to treat the analysis of MD data as a learning task with the hope of recognizing dynamic patterns that can provide new biological insights for this field, thus interpretability is a basic requirement for method design.
>
>
> [1] Molecular dynamics and protein function. PANS, 2005.
>
> [2] Dynamic personalities of proteins. Nature, 2007.
>
> [3] Ab initio characterization of protein molecular dynamics with AI2BMD. Nature, 2024.
>
>
> **Q2:** I do not understand your setting in the EGFR dynamics experiment since I'm not an expert on this domain. Could you please explain why the trajectory can be labeled according to its sensitivity towards the drug? I think the sensitivity should be a number rather than a 0/1 label.
>
>
> **Response:** We apologize for the confusion. In this context, drug sensitivity refers to whether a drug can bind to a protein so as to exert its function, therefore it is typically categorized as either bound (1) or not bound (0).

---

> > ### Comment · Reviewer_9koh · 2024-11-25
> >
> > Thank you for your comprehensive responses. I have thoroughly reviewed them and concur with the authors' insights. Given the limited availability of datasets, gathering more MD simulation data is indeed a significant challenge. Furthermore, your additional experiments underscore the necessity of your approaches in addressing the challenges faced in large-scale molecular systems. I appreciate the authors' detailed response and am pleased to increase my score.

---

> > > ### Author Response · Authors · 2024-11-25
> > > **Thank you so much!**
> > >
> > > Dear Reviewer,
> > >
> > > Thank you for your recognition and valuable feedback! As you rightly pointed out, there are indeed many issues that remain to be addressed in this field, and it can be challenging to tackle all of them in a single study. We believe that our current work represents a foundational step for future investigations into molecular dynamics. We sincerely appreciate the time and effort you dedicated to reviewing our research.

---

### Official Review · Reviewer_diJk · 2024-11-03

**Soundness:** 3
**Presentation:** 3
**Contribution:** 2
**Rating:** 8
**Confidence:** 3

**Summary:**

The paper introduces Deep Signature, a framework for characterizing dynamics on graphs. Deep Signature involves three key components: (1) graph coarsening using a deep spectral clustering module, (2) computation of path signatures to capture global inter-node interactions using iterated integrals over time, and (3) a two-layer MLP for property prediction.
The authors tested Deep Signature on three datasets: (1) gene expression dynamics, classifying into degradation or dimerization types, (2) GPCR dynamics, distinguishing between active and inactive states, and (3) EGFR mutant dynamics, predicting drug sensitivity. They conducted an ablation study validating the contribution of each loss component in the model and performed limited comparisons with methods based on static structures, dihedral angles, and GraphLSTM.

**Strengths:**

1. Incorporates appropriate symmetries for (temporal) 3D point cloud learning. In particular, the invariance to time reparameterization is key because the underlying MD simulations might be prone to random restarts and miscellaneous artefacts preventing them from being a smooth "video" (ie, the MD itself might be erratic with the same state sampled repeatedly).

2. Deep Signature _learns_ the ideal CG beads relevant to the task, bypassing the need to manually remove degrees of freedom (eg: CA-level coarse-graining, as done in existing methods). The use of the signature transform allows for local and global temporal interactions to be captured well, as opposed to learning representations on separate frames.

3. I'm confident this method can be used for good representation learning of trajectory information for tasks beyond the ones mentioned in the paper (eg: interpolation in the latent space of protein conformations, perhaps for ensemble generation).

4. Demonstrates relatively good performance compared to weak and strong temporal graph learning baselines (like GraphLSTM).

**Weaknesses:**

The paper does not clarify the level of coarsening necessary to make path signature computations feasible. For instance, in gene regulatory dynamics, the graph was reduced from 100 nodes to 30, while the EGFR dataset was reduced from approximately 5000 atoms to 50 nodes, but the level of coarsening for the GPCR dataset is unspecified. This raises questions about whether 50 nodes is a computational limit for the method. Additionally, while protein structure can be intuitively coarsened into backbones, sidechains, and motifs, the interpretation for coarsened graphs in gene expression dynamics is unclear. It is uncertain if the coarsened nodes correspond to gene hubs or other biologically meaningful groupings.

The equation used to model GRN dynamics (Eqn 13 in Section B.1) appears incorrect. Specifically, for dimerization (when f=2), the concentration should should be squared under Michaelis-Menten kinetics, rather than simply doubling the decay rate. This could lead to inaccurate modeling of gene expression dynamics and affect the results in this section.

The comparisons used in the experiments are limited and static, primarily involving the first and last frames (head, tail, and head & tail) which do not capture temporal dynamics. The authors do not benchmark against dynamic approaches that consider time-varying information, such as MDTraj [1], Timewarp [2], and DSR [3]. Including these comparisons would provide a more rigorous evaluation of Deep Signature's effectiveness relative to established tools for molecular trajectory analysis.

The authors claim that the coarsened dynamics in Fig 6c follow the same trend as the original dynamics, yet this similarity is not quantified. Providing a quantitative measure, such as correlation coefficients between the original and coarsened dynamics at various coarsening levels, would better support this claim. Additionally, the paper would benefit from comparisons of the authors' coarsening strategy against other deterministic and learnable methods for protein graph coarsening to demonstrate its effectiveness and fidelity in preserving dynamics.

The description of the cross-validation process and test set creation is confusing. The authors mention “for each running, we evaluate the prediction accuracy of our method on an independent unseen test set and report the averaged results,” but it is not clear how this set is constructed. If results are averaged, please report the standard deviation in all the tables.
[1] MDTraj: A Modern Open Library for the Analysis of Molecular Dynamics Trajectories. Biophysical Journal, 2015.
[2] Timewarp: Transferable Acceleration of Molecular Dynamics by Learning Time-Coarsened Dynamics. NeurIPS 2023
[3] DSR: Dynamical Surface Representation as Implicit Neural Networks for Protein. NeurIPS 2023.

**Questions:**

1. Existing equivariant neural networks with geometric inductive biases have been outperformed, both in terms of performance and efficiency, by Transformer-style architectures (ie, something as simple as torch.nn.TransformerEncoder). Was any ablation done that compares Deep Signature with such architectures that "tokenize" the trajectory intervals?

2. I'm curious what the learned coarse-grained beads/maps look like for GPCR proteins – is it creating CG beads only based on atoms within a locality? How does this compare to naively taking atoms within some radial ball and considering them a bead? More often than not, these naive CG choices work well in practice, without the need to overcomplicate it with a learnable method. I'd like to see whether this "learned GCN-based coarse-graining" is really necessary.

3. How sensitive is the path signature transform to very dynamic conformational changes in a short duration of time? For instance, as I mentioned above, fast-folding proteins undergo significant changes in 3D shape in just short simulations. Can this method capture this dynamics well enough?

---

> ### Author Response · Authors · 2024-11-22
> **Thank you; Address your concerns [1/2]**
>
> Dear Reviewer,
>
> Thank you for dedicating your time and effort to reviewing our paper. We are glad that you recognise our method as well-motivated and practical value in broad applications (such as ensemble generation). We took your suggestions seriously, and have conducted additional experiments to update our paper. Please see below for our point-to-point responses.
>
> ---
>
> **W1:** The paper does not clarify the level of coarsening necessary to make path signature computations feasible…but the level of coarsening for the GPCR dataset is unspecified.
>
> **Response:** Thank you for your suggestion. We specified the coarsening level for GPCR as ``first coarsen the dynamics into 100 nodes and subsequently into 50 nodes’’ in B.3 Section of the Appendix to save space. Currently, the coarsening level is determined in a heuristic and empirically driven manner by trading off between the granularity of coarsening and the complexity of signature transform. However, our experiments on gene regulatory dynamics, as shown in Fig. 4 (b), demonstrate that our method is highly robust across various system scales, even when using a fixed coarsening level.
>
> To further validate this robustness, we conducted additional experiments by varying the coarsening levels for the EGFR and GPCR datasets. The results are summarized below:
>
> |Coarsening Level| 100| 50| 30| 10| 5|
> |:---:|:---:|:---:|:---:|:---:|:---:|
> |EGFR Accuracy (%)| 66.93 ± 4.62 | 69.33 ± 4.78 | 66.40 ± 5.06 | 63.20±7.56   | 57.33±4.73 |
> |GPCR Accuracy (%)| 50.67 ± 5.58 | 58.00 ± 4.17 | 58.00 ± 6.01 | 57.73 ± 8.92 | 55.07 ± 6.32 |
>
> As we can see, both 30 and 50 can lead to desired results, showing the robustness of our method across different coarsening levels.
>
> **W2:** The equation used to model GRN dynamics (Eqn 13 in Section B.1) appears incorrect.
>
> **Response:** Thank you for kindly pointing out our writing error, we have revised it in our new submission. Our implementation for generating gene regulatory dynamics is based on the code provided by [1], where $f$ is correctly squared.
>
> [1] Neural Dynamics on Complex Networks. KDD, 2020.
>
> **W3:** The comparisons used in the experiments are limited and static, primarily involving the first and last frames (head, tail, and head & tail) which do not capture temporal dynamics.
>
> **Response:** Thank you for providing the references. The comparison is limited mainly because no existing techniques can process the high spatial and temporal complexity of large-scale molecular dynamics, and our work is the first to approach this challenge. MDTraj [1] was employed by our baselines `Dihedral angle` and `Cα-dihedral angle` to compute the dihedral angles, which were then combined with PCA and SVM from scikit-learn to enable classification. Timewarp [2] is an enhanced sampling approach designed to explore transitions between metastable states, thus can be integrated into our framework as a preprocessing operation to reduce temporal complexity rather than as a standalone method for characterizing dynamics into representations. Lastly, DSR [3] represents dynamic protein surfaces via a time-varying distance function that estimates the minimum distance from a point to the surface but does not provide explicit representations, making it unsuitable for our task. We include a discussion of this difference in Line 122-123 of our revised version.
>
> **W4:** The authors claim that the coarsened dynamics in Fig 6c follow the same trend as the original dynamics, yet this similarity is not quantified.
>
> **Response:** Thank you for your insightful suggestion. We present qualitative results to demonstrate that our method is capable of preserving the original dynamics, though this alone is not sufficient to guarantee the best classification performance. In our method, coarse graining is tightly coupled with the surrogate task, this means our method learns to aggregate dynamics in a way that optimally benefits the classification performance through the **learnable linear aggregation** $\mathbf{Q}$ in Eq. (2). This differs from conventional coarse graining techniques that typically focus on extracting stable substructures, such as motifs, and then **averaging** their dynamics to best preserve the original system’s behavior.
>
>
> To verify the difference in learning behavior,  we vary the coefficient $\lambda_2$ which adjusts the respecting level for original dynamics and the results on EGFR dataset are given below:
> |$\lambda_2$| 10.0| 1.0| 0.1| 0.01|
> |:---:|:---:|:---:|:---:|:---:|
> | Accuracy (%)  | 65.27 ± 2.48 | 67.67 ± 1.63 | 67.80 ± 1.15 | 69.33 ± 4.78 |
>
> As seen, increasing the emphasis on preserving the original system's behavior does not necessarily lead to improved performance. This highlights the importance of balancing dynamic preservation with the surrogate task to achieve the best results.

---

> ### Author Response · Authors · 2024-11-22
> **Thank you; Address your concerns [2/2]**
>
> **W5:** The description of the cross-validation process and test set creation is confusing.
>
>
> **Response:** We apologize for the confusion. This process involves partitioning long MD simulations into individual data points as described in [2]. In our experiments, we ensure that the MD simulations used to generate the test set remain disjoint from those in the training and validation set. This is a more realistic yet challenging setup that requires the algorithm to generalize to heterogeneous distribution where test data may exhibit unseen structures and varying numbers of nodes. We adopt the suggestion of reporting the standard deviation, and have revised our manuscript accordingly.
>
>
> [2] An Interpretable Convolutional Neural Network Framework for Analyzing Molecular Dynamics Trajectories: a Case Study on Functional States for G‑Protein-Coupled Receptors. Journal of Chemical Information and Modeling, 2022.
>
>
> **Q1:** Existing equivariant neural networks … Was any ablation done that compares Deep Signature with such architectures that "tokenize" the trajectory intervals?
>
>
> **Response:** Thank you for the valuable question. We have conducted addition comparison experiments between path signature transform and Transformer on EGFR and GPCR datasets, and the results are listed below:
> | Model                   | **EGFR Accuracy (%)** | **EGFR Recall (%)** | **GPCR Accuracy (%)** | **GPCR Recall (%)** |
> |-------------------------|-----------------------|----------------------|-----------------------|----------------------|
> | Transformer             | 64.40±6.55         | 20.80±10.47         | 44.80 ± 6.97         | 33.33 ± 21.08        |
> | Path Signature Transform| 69.33 ± 4.78         | 21.27 ± 8.26         | 58.00 ± 4.17         | 43.30 ± 7.85         |
>
>
> We conjecture the reason that Transformer is not well-suited for these tasks due to (1) it degrades spatial correlations, which are crucial for deciphering interatomic interactions; and (2) it lacks the capability to effectively process long-range simulations, as demonstrated in [3].
>
>
> [3] Big Bird: Transformers for Longer Sequences. NeurIPS, 2020.
>
>
> **Q2:** I'm curious what the learned coarse-grained beads/maps look like for GPCR proteins – is it creating CG beads only based on atoms within a locality? How does this compare to naively taking atoms within some radial ball and considering them a bead?
>
>
> **Response:** Thank you for your question. Our spectral clustering module relies on the local connectivity of covalent bonds, thus it naturally aggregates local atoms to create CG beads, and locality is a fundamental and valid requirement for this process. However, what makes our method different from conventional coarse graining techniques is that some CG beads may represent a significantly larger number of atoms that are not physically close. This can be verified by our Fig. 6.
>
>
> We also performed a comparison experiment using radial ball for coarse graining the GPCR dataset. The results are presented below:
> | Method          | Radial Ball (0.5A) | Radial Ball (1.0A) | Radial Ball (1.5A) | Deep Signature |
> |------------------|:------------------:|:------------------:|:------------------:|:--------------:|
> | Number of Beads | ~150              | ~50               | ~20               | 50             |
> | Accuracy (%)    | 56.40 ± 3.81      | 53.60 ± 2.37      | 45.07 ± 3.34      | 58.00 ± 4.17   |
>
>
> It should be noted that, coarse graining according to the radial ball results in  inconsistent numbers of beads across different molecules. While we can align them by truncation, it would compromise generalization capability.
>
>
> **Q3:** How sensitive is the path signature transform to very dynamic conformational changes in a short duration of time?
>
>
> **Response:** Thank you for the insightful question. Path signature is inherently sensitive to dynamic conformational changes, even over short durations, due to its use of the integral operator. With this operation, the fine-grained dynamic information can be captured and encoded [4, 5]. Additionally, the hierarchical nature of the path signature transform enables it to represent higher-order interactions and complex dependencies over time, making it particularly suitable for tasks involving highly dynamic and nonlinear systems, such as molecular dynamics.
>
>
> [4] A Primer on the Signature Method in Machine Learning. ArXiv, 2016.
>
> [5] Signature Moments to Characterize Laws of Stochastic Processes. JMLR, 2022.

---

> > ### Comment · Reviewer_diJk · 2024-11-28
> > **Author efforts in additional experimentation**
> >
> > I appreciate the authors effort in additional experimentation regarding coarse graining and dynamics preservation quantification. Therefore, I am happy to increase my score.

---

> > > ### Author Response · Authors · 2024-11-28
> > > **Thank you so much!**
> > >
> > > We are glad to hear that our responses help address your concern. Your valuable comments and suggestions have greatly enhanced the quality of our paper. Thank you so much for your time and effort to review our work!

---

### Meta-Review · Area_Chair_e6Vp · 2024-12-18

**Metareview:**

This paper introduces Deep Signature, a novel and computationally efficient framework for characterizing complex dynamics and interatomic interactions based on their evolving trajectories. The authors propose an approach that combines soft spectral clustering for local aggregation of cooperative dynamics and a signature transform that collects iterated integrals to provide a global representation of non-smooth interactive dynamics. Theoretical analysis establishes that Deep Signature possesses several desirable properties. The experimental evaluation on three benchmark biological processes demonstrates that the proposed framework outperforms baseline methods, highlighting its potential for applications in dynamic systems analysis.

Strengths:

1. The method is presented in a clear and easily understandable manner.

2. The proposed approach is well-motivated and thoughtfully designed, incorporating appropriate symmetries, learning relevant coarse-grained (CG) beads for the task, and requiring no expert knowledge.

3. The framework’s desirable properties are backed by solid theoretical analysis.

4. The method demonstrates strong potential for effective representation learning of trajectory information, with competitive performance relative to both weak and strong temporal graph learning baselines.

Weaknesses:

1. There is a lack of experiments conducted on large datasets.

2. The experimental evaluation and comparison to baseline methods remain insufficient.

3. Several typographical errors and unclear presentations need attention.

Overall, the proposed method is novel and offers significant value to the molecular dynamics (MD) field, with desirable performance that suggests its potential for impactful downstream applications. After reviewing the rebuttal, all reviewers provided positive assessments. Therefore, I recommend acceptance.

**Additional Comments On Reviewer Discussion:**

During the rebuttal period, the authors addressed the following points:

Comparison with Existing Coarse-Graining Methods: Reviewers diJk and xZWn requested comparisons with existing coarse-graining methods. In response, the authors conducted additional experiments, including a comparison using a radial ball method for coarse-graining the GPCR dataset and further experiments varying the coarsening levels for both the EGFR and GPCR datasets. Both reviewers expressed satisfaction with these additional experiments.

Additional Ablation Studies: Reviewers diJk and xZWn also requested more detailed ablation studies. The authors responded by performing extra experiments to investigate the effects of the number of beads, loss coefficient parameters, and the number of coarse-grained (CG) beads.

Concerns About Limited and Static Comparisons: Reviewer diJk raised concerns about the limited and static nature of the comparisons in the experiments. The authors clarified the complexity of the task and emphasized that they are the first to tackle this challenge, providing additional context to justify the scope of their experimental comparisons.

Overall, the authors have provided thorough explanations and supplementary results that adequately address all reviewers' concerns. During the author-reviewer discussion period, all reviewers revised their assessments to positive scores.

---

### Decision · Program_Chairs · 2025-01-22

Accept (Poster)